

# CD4+ and CD8+ cell counts are significantly correlated with absolute lymphocyte count in hospitalized COVID-19 patients: a retrospective study

Phey Liana[1], Aprilia Paskah Samosir[2], Nurmalia Purnama Sari[1], Raden Ayu Linda Andriani[3], Verdiansah Verdiansah[1], Hidayatullah Hidayatullah[3], Zen Ahmad[3] and Tungki Pratama Umar[2]

[1] Department of Clinical Pathology, Faculty of Medicine, Universitas Sriwijaya, Palembang, South Sumatera, Indonesia
[2] Faculty of Medicine, Universitas Sriwijaya, Palembang, South Sumatera, Indonesia
[3] Department of Internal Medicine, Faculty of Medicine, Universitas Sriwijaya, Palembang, South Sumatera, Indonesia

Corresponding author
Phey Liana, pheyliana@fk.unsri.ac.id

## ABSTRACT

**Background:** Coronavirus disease 2019 (COVID-19) is a contagious respiratory illness that was declared a pandemic in March 2020. Lymphopenia is one of the specific laboratory results disturbance in COVID-19 patients. Such findings are frequently associated with substantial changes in T-cell counts, particularly CD4+ and CD8+ T-cells. This study aimed to examine the correlation between CD4+ and CD8+ cell counts and absolute lymphocyte count (ALC) in COVID-19 patients and analyze its difference based on the COVID-19 patients' severity.

**Methods:** From March 2022 to May 2022, we conducted a retrospective cohort study using medical records and laboratory data from patients diagnosed with COVID-19 at our hospital who met the inclusion and exclusion criteria. The total sampling method was used to recruit study participants. We conducted bivariate analysis, which consisted of correlation and comparative analysis.

**Results:** Thirty-five patients met the inclusion and exclusion criteria and were divided into two severity groups (mild-moderate and severe-critical). The findings of this study revealed a significant correlation between CD4+ cell count and ALC on admission (r = 0.69, p < 0.001) and the tenth day of onset (r = 0.559, p < 0.001). Similarly, there was a correlation between CD8+ and ALC at admission (r = 0.543, p = 0.001) and on the tenth day of onset (r = 0.532, p = 0.001). Individuals with severe-critical illness had lower ALC, CD4+, and CD8+ cell counts than those with mild-moderate illness.

**Conclusion:** According to the findings of this study, there is a correlation between CD4+ and CD8+ cell counts and ALC in COVID-19 patients. All lymphocyte subsets also showed a lower value in severe forms of the disease.

## INTRODUCTION

Coronavirus disease 2019 (COVID-19) is an infectious disease caused by the Severe Acute Respiratory Syndrome Coronavirus 2 (SARS-CoV-2). COVID-19 was declared a pandemic in March 2020 and still affects the global community (*Jain et al., 2022*). Until March 2023, COVID-19 has caused over 750 million confirmed cases and 6.8 million deaths (*World Health Organization, 2023*). Besides the high infectivity of the disease, some people developed a long COVID-19 phenomenon, which significantly impacts their lives due to prolonged complaints, including dyspnea, fatigue, and sleep disturbance while also developing many laboratory parameters abnormalities (*Davis et al., 2023*). Furthermore, laboratory parameters have been described as an important predictor of COVID-19 severity and mortality, particularly the hematology examination of lymphocytes (*Marwah et al., 2021*).

Lymphopenia, a decrease in absolute lymphocyte count (ALC), is one of the most common laboratory disturbances seen in COVID-19 patients (*Umar & Siburian, 2022*). This specific finding is highly related to inflammatory cytokine storm (characterized by an elevation of pro-inflammatory cytokines, such as Tumor necrosis factor-alpha/TNF-α and interleukin (IL)-6), exhaustion of T cells, and direct SARS-CoV-2 infection to the T-cells (*Tavakolpour et al., 2020*). Lymphopenia can influence the prognosis of COVID-19 patients because it is a systemic manifestation of angiotensin-converting enzyme 2 (ACE2) receptors overexpression on the surface of lymphocytes and T-cells. T-cells themselves are a vital component of the adaptive immune response to viral infections. CD8+ T cells are important because of their specific cytotoxicity against infected cells, whereas CD4+ T cells are essential because they are supporting CD8+ and B cell activation while regulating cytokines production (*Moss, 2022*).

In a previous investigation, severe COVID-19 patients had lower lymphocyte subsets level than mild cases (*Zhang et al., 2020*). The researchers discovered lymphopenia in severe COVID-19 patients was associated with decreased T cell counts, specifically CD4+ and CD8+ (*Wang et al., 2020*). Furthermore, a study discovered that deceased COVID-19 participants had markedly decreased total lymphocytes, CD3+ T cell, CD4+ T cell, CD8+ T cell, and CD19+ B cell counts than COVID-19 patients who survived (*Cantenys-Molina et al., 2021*).

Although CD4+ and CD8+ significantly impact COVID-19 severity and mortality, these parameters are expensive and, in some places, inaccessible. As an alternative, ALC, a simple, cost-effective, and widely available examination is routinely performed (*Kumarasamy et al., 2002*; *Mahajan et al., 2004*). Currently, studies examining the relationship between ALC, CD4+, and CD8+ have yet to be conducted in Indonesia. Thus, we intend to investigate the correlation between CD4+ and CD8+ cell counts and ALC in a tertiary hospital setting. In addition, we also examined lymphocyte subsets level and analyze its difference based on COVID-19 patients' severity.

## MATERIALS AND METHODS

This retrospective cohort study utilized the medical records and laboratory data of hospitalized adults (≥18 years) with COVID-19 at Dr. Mohammad Hoesin Hospital,

Palembang, between March and May 2022. COVID-19 status was confirmed by a positive reverse-transcriptase polymerase chain reaction (RT-PCR) test. Patients were selected through a total sampling procedure and followed since admission for 10 days. This secondary time point was selected because CD4+ and CD8+ formation at the acute phase of the disease reached its peak on the tenth day of COVID-19 onset (*Stephens & McElrath, 2020*). Patients with autoimmune disorders, immunodeficiency, cancer, and tuberculosis were excluded. Consent was not required due to the retrospective nature of this study. The Sriwijaya University Faculty of Medicine Ethics Committee approved the study procedure (Approval number: 212-2022).

The Sysmex XN-1000 was used for hematology testing, including complete blood count and ALC. The BD FACSLyric Flow cytometry was used to count CD4+ and CD8+. The serum specimen was collected twice, once on admission and once on the tenth day of onset.

Descriptive data of confirmed COVID-19 patients (age, gender, and laboratory examination) were described using univariate analysis. Normality testing was done by using the Shapiro-Wilk test to determine data distribution and appropriate statistical tests. Bivariate analysis was used to determine the correlation between CD4+ and CD8+ cell counts and ALC using the Spearman test. Correlation strength was determined based on the previous classification from *Chan (2003)*. Furthermore, we also analyzed the difference of each parameter using the independent T-test (normal distribution) or Mann-Whitney U test (non-normal distribution) based on patient severity (for unpaired data). Meanwhile, the Wilcoxon signed-rant test was utilized to determine the significance of lymphocyte subsets dynamic (paired data). We used SPSS Statistics for Windows (Version 26.0., Armonk, NY: IBM Corp.) and GraphPad Prism (Version 9.5.1., San Diego, CA: GraphPad Software) to undertake statistical analysis. The *p*-value at <0.05 is considered significant.

## RESULTS

This study included 35 patients. They were classified as mild to moderate (15 patients) or severe to critical (20 patients) severity (Table 1). The average age of all study participants was $52.69 \pm 15.70$ years, with no differences between groups ($p = 0.820$). There was also no discrepancy based on the gender of enrolled participants ($p = 1.000$). Because the data did not follow normal distribution, erythrocyte, lymphocyte, and neutrophil were analyzed using the Mann-Whitney U test. Whereas, other parameters (hemoglobin, hematocrit, and leukocyte) were analyzed using Independent T-test (normal distribution). Based on the patients' severity, lymphocyte value was significantly lower in severe-critical patients than those with mild-moderate conditions at admission ($p = 0.002$) and on the tenth day of onset ($p < 0.001$). Meanwhile, substantially higher values were observed at the admission and tenth day of onset for leukocyte ($p = 0.040$, $p = 0.011$, respectively) and neutrophil ($p = 0.003$, $p < 0.001$, respectively). Furthermore, ALC levels, CD4+, and CD8+ cell counts tended to be lower in patients with severe-critical symptoms than in patients with mild-moderate symptoms on admission ($p = 0.016$, $p = 0.013$, $p = 0.003$, respectively) and tenth day of onset ($p = 0.002$, $p < 0.001$, $p = 0.002$, respectively).

**Table 1 Characteristics of COVID-19 patients.**

| No | Parameter | Total ($n$ = 35) | Symptom | | $p$-value |
|----|-----------|------------------|---------|---|-----------|
| | | | **Mild-moderate ($n$ = 15)** | **Severe-critical ($n$ = 20)** | |
| 1 | Age (years) | 52.69 ± 15.70 | 53.40 ± 17.17 | 52.15 ± 14.94 | 0.820[a] |
| 2 | Gender | | | | |
| | • Male | 21 (60%) | 9 (60%) | 12 (60%) | 1.000[c] |
| | • Female | 14 (40%) | 6 (40%) | 8 (40%) | |
| 3 | Hematology examination | | | | |
| | a. Hemoglobin (g/dL) | | | | |
| | • On admission | 10.04 ± 2.299 | 9.92 ± 2.494 | 10.14 ± 2.203 | 0.784[a] |
| | • Tenth day | 10.03 ± 1.908 | 10.16 ± 2.056 | 9.93 ± 1.838 | 0.736[a] |
| | b. Hematocrit (%) | | | | |
| | • On admission | 30.40 ± 6.822 | 30.40 ± 7.491 | 30.40 ± 6.476 | 1.000[a] |
| | • Tenth day | 30.17 ± 5.602 | 30.80 ± 5.697 | 29.70 ± 5.630 | 0.573[a] |
| | c. Erythrocyte ($\times 10^6$/mm$^3$) | | | | |
| | • On admission | 3.7 (1.89–24) | 3.77 (1.89–24) | 3.64 (1.95–5.38) | 0.400[b] |
| | • Tenth day | 3.72 (2.17–22) | 3.77 (2.17–22) | 3.54 (2.48–4.67) | 0.364[b] |
| | d. Leukocyte ($\times 10^3$/mm$^3$) | | | | |
| | • On admission | 12.706 ± 6.699 | 10.04 ± 5.673 | 14.7 ± 6.841 | 0.040[a] |
| | • Tenth day | 10.92 (3.7–26.1) | 7.92 (3.7–26.07) | 11.92 (7.48–26.1) | 0.011[a] |
| | e. Lymphocyte (%) | | | | |
| | • On admission | 10 (1–49) | 16 (4–49) | 6.5 (1–23) | 0.002[b] |
| | • Tenth day | 9 (1–51) | 17 (4–51) | 7 (1–17) | <0.001[b] |
| | f. Neutrophil (%) | | | | |
| | • On admission | 81 (40–95) | 70 (40–94) | 89 (57–95) | 0.003[b] |
| | • Tenth day | 83 (37–95) | 70 (37–93) | 88 (65–95) | <0.001[b] |
| 4 | ALC (/mm$^3$) | 1,117.6 (218.2–4,578.1) | 1,332 (718.4–2,886.1) | 1,020.5 (218.2–4,578.1) | 0.016[b] |
| | • On admission | 1,039.8 (261–5,467.2) | 1,361 (541.6–5,467.2) | 879.5 (261–2,483) | 0.002[b] |
| | • Tenth day | | | | |
| 5 | CD4+ cell count (/mm$^3$) | | | | |
| | • On admission | 372 (3–2,810) | 734 (3–2,810) | 307 (5–909) | 0.013[b] |
| | • Tenth day | 342 (3–2,103) | 817 (26–2,103) | 195 (3–779) | <0.001[b] |
| 6 | CD8+ cell count (/mm$^3$) | | | | |
| | • On admission | 279 (17–1,361) | 412 (17–1,361) | 183.5 (70–791) | 0.003[b] |
| | • Tenth day | 255 (6–959) | 485 (120–959) | 178 (6–566) | 0.002[b] |

**Note:**
Data is presented as n (%), median (minimum–maximum), or mean ± standard deviation. $p$-values were calculated by [a]Mann–Whitney test, [b]Independent T-test, or [c]Chi-square test. ALC, Absolute Lymphocyte Count.

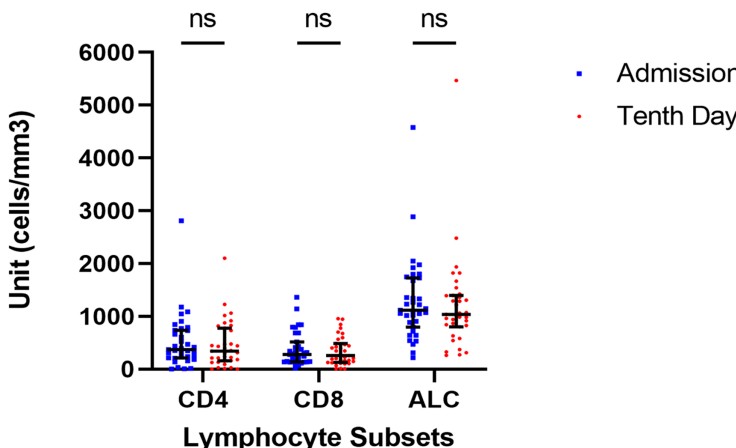

**Figure 1 Dynamic of lymphocyte subsets.** There are no significant changes in any lymphocyte subset from the day of admission to the tenth day of admission (Wilcoxon test, $p > 0.05$).

We observed the dynamics of lymphocyte subsets in our patient cohort. However, all parameters (CD4+, CD8+, and ALC) did not show any significant difference during the observation period based on the Wilcoxon signed-rank test ($p = 0.288$; $p = 0.383$; $p = 0,288$; respectively) from the day of admission to the tenth day of onset which was done because the data did not follow the normal distribution. The data were presented in Fig. 1. Similar findings were also found after we analyzed study participants based on their severity groups (Fig. S1).

Correlational analysis (Fig. 2) showed that ALC is significantly correlated with CD4+ and CD8+ counts at admission and on the tenth day after illness onset. A significant and moderately strong correlation ($r = 0.69$, $p < 0.001$) was found between CD4+ count and ALC on the day of admission. Meanwhile, on the tenth day of onset, there was a fair correlation ($r = 0.559$, $p < 0.001$) between the number of CD4+ counts and ALC. Furthermore, CD8+ count and ALC have a fair correlation both at the day of admission ($r = 0.543$, $p = 0.001$) and the tenth day of disease onset ($r = 0.532$, $p = 0.001$).

## DISCUSSION

The hematological assessment showed a noticeable difference between all leukocyte parameters observed in this research (leukocyte, lymphocyte, and neutrophil). This aligns with a prior research conducted in a similar population during the earlier phase of the COVID-19 pandemic (*Hilda et al., 2022*). This disruption is linked to the interaction between ACE2 receptors and SARS-CoV-2, which causes severe inflammation (*Medina-Enríquez et al., 2020*). Furthermore, COVID-19 caused excessive production of proinflammatory cytokines such as IL-6, IL-2, IL-7, granulocyte-colony stimulating factor (G-CSF), Macrophage Inflammatory Protein-1 Alpha ((MIP-1-α)/CCL3, and TNF-α), termed as cytokine storm (*Chen et al., 2021*). This is also manifested as a disruption in leukocyte-related parameters.

This study discovered that the severe-critical group had significantly lower CD4+ and CD8+ values than the mild-moderate group. A similar finding was discovered in China

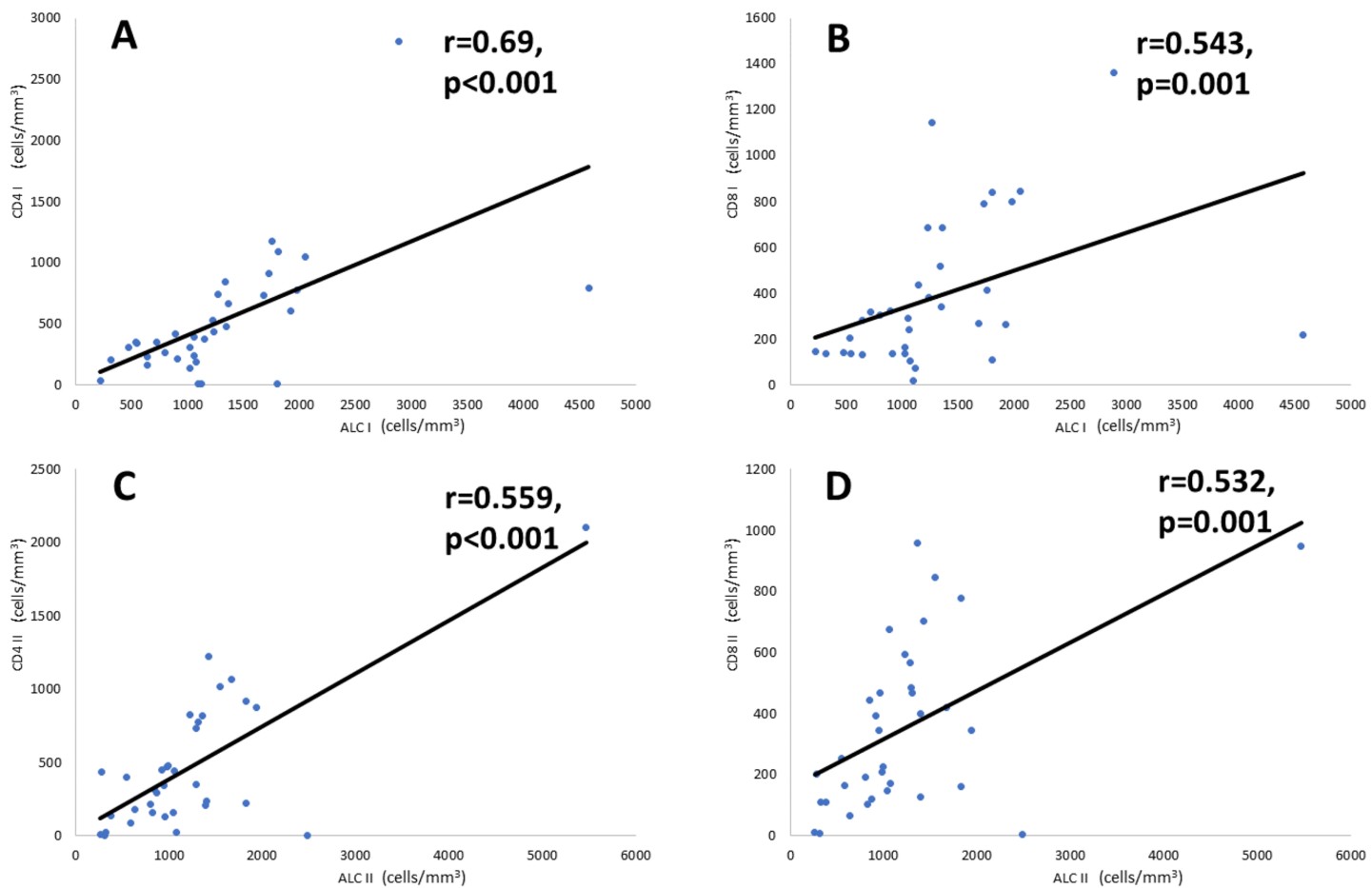

**Figure 2 Correlation of CD4+ and CD8+ cell counts with ALC.** Correlation analysis is performed using Spearman's rank correlation coefficient. (A) CD4+ and ALC on the day of admission, (B) CD8+ and ALC on the day of admission, (C) CD4+ and ALC on the tenth day of onset, and (D) CD8+ and ALC on the tenth day of onset. Note: ALC, Absolute Lymphocyte Count.

where COVID-19 patients with severe disease activity had a significant reduction of CD4+ and CD8+ cell count ($p < 0.05$) compared with patients' with mild-moderate severity (*Jiang et al., 2020*). *Sun et al. (2020)*, who collected data from 63 study participants (19 patients with severe-critical status), found that individuals with severe and critical symptoms had low CD4+ and CD8+ cell counts. A previous study has shown a significant association between decreased CD4+ cell count and patient's clinical deterioration during hospitalization (*Calvet et al., 2020*). Physiologically, CD4+ cells contributed to the immune response through the cytokine secretion process. CD4+ cells have many functions, such as regulating activation of the innate immune system, B lymphocytes, and CD8+ T cells while suppressing the body's immune reactions to prevent a hyperinflammatory state. Ineffective CD4+ and CD8+ functions will result in a massive degree of inflammation, which can severely exacerbate the clinical state (*Caldrer et al., 2021*; *de Candia et al., 2021*). The differences, as presented in an examination of CD4+ kinetics, were consistent across the disease course, from the beginning to the fourth week of COVID-19 onset (*Koblischke et al., 2020*).

In COVID-19 patients, lymphopenia is denoted as an important laboratory parameter disturbance. This is consistent with the findings of our study, which show a lower ALC in patients with severe-critical status than those with milder disease conditions. Furthermore, lymphopenia was observed in previous studies and determined to be significantly related to the overall disease condition (*Amin et al., 2021*; *Tan et al., 2020*). Lymphopenia is linked to the body's proinflammatory state during SARS-CoV-2 infection, associated with direct lymphocyte infection by the virus or immunological apoptosis of lymphocytes (particularly during cytokine storm). Lymphopenia can also be induced by the direct impact of the SARS-CoV-2 to suppress bone marrow activity (*Demoliou, Papaneophytou & Nicolaidou, 2022*; *Guo et al., 2021*).

A decrease in the number of CD4+ and CD8+ cells also characterizes lymphopenia. Lymphopenia can inhibit virus clearance and trigger prolonged inflammation, ultimately causing organ damage. COVID-19 patients with mild-moderate severity who recovered quickly have a statistically higher amount of T cells, CD4+ cells, and CD8+ cells (*Liu et al., 2020*; *Ramljak et al., 2021*). However, from the time of admission until the tenth day of onset, our investigation did not detect any statistically significant shifts in lymphocyte subsets. No comparable findings based on the severity of the illness were found in earlier research. Meanwhile, *Rezaei et al. (2021)* did not discover any significantly different deviations in CD4+ and CD8+ dynamics in deceased COVID-19 patients after 7 days of study.

Our research examined the correlation between ALC, CD4+ cells, and CD8+ cells in COVID-19 patients. According to our exploration, this assessment is still scarce in the literature. Our study supports the finding of a previous study, which discovered a strong correlation coefficient between CD4+ in peripheral blood and total lymphocytes in COVID-19 patients ($r = 0.9051$, $p < 0.01$) (*Sun et al., 2020*). Meanwhile, prior research on Human Immunodeficiency Virus/Acquired Immune Deficiency Syndrome (HIV/AIDS) patients identified a significant relationship between ALC and CD4 cell count with an r-value of 0.327 ($p < 0.05$) (*Agrawal, Rane & Jadhav, 2016*). Nevertheless, in the multiple sclerosis patients, both CD4+ and CD8+ T-cells correlated significantly with ALC from treatment initiation to week 96 of observation ($r = 0.559–0.880$; $p < 0.001$) (*Longbrake et al., 2021*). Studies have found that the absolute number of T-lymphocytes, CD4+ T-cells, and CD8+ T-cells in COVID-19 patients with severe-critical condition is lower than in patients with mild-moderate condition, hypothesized as an impact of direct SARS-CoV-2 infection to T-lymphocytes, specifically CD4+ and CD8+ T-cells (*Shen et al., 2022*; *Wen et al., 2021*). Furthermore, the aforementioned association is not surprising given that the adaptive immune system, primarily composed of lymphocyte T cells, is divided into two main classes, CD4+ and CD8+, based on the expression of an accessory glycoprotein co-receptor, which is liable for their interplay with major histocompatibility complex (MHC) class II or class I, respectively (*Kumar, Connors & Farber, 2018*). Researchers have proposed replacing or substituting one test (of the lymphocyte subset) with another due to the moderate or strong correlation between the tests. However, its application must be accompanied by a rigorous investigation of content and construct validity while undertaking re-testing on similar purposes for a specific study population (*Sadeghi, 2013*).

The current study has some limitations. Our sample size is small, limiting generalizability; thus, it can be considered a pilot study. Then, there is a discrepancy in the timing of the examination of ALC, CD4+, and CD8+ cell counts. Furthermore, because ALC data is scarce on the tenth day of admission, some samples cannot be enrolled in this research.

## CONCLUSIONS

There was a moderately strong correlation between ALC and CD4+ on patient entry and a fair correlation between CD4+ on the tenth day of disease onset and CD8+ over the entire study period in our cohort. Thus, the ALC could be a surrogate marker for CD4+ at early COVID-19 stage in low-resource settings. We also found significant differences in hematological parameters (leukocyte, neutrophil, and lymphocyte) between severity classes. For the specific examination of lymphocyte subsets, all metrics were substantially lower in severe-critical status patients compared to mild-moderate status patients at admission and on the tenth day of onset. Future studies should investigate this finding in a larger patient population and determine the consistency of CD4+ T-cells, CD8+ T-cells, and ALC correlation and confirming T-cell exhaustion and apoptosis, especially during severe COVID-19 state. Furthermore, CD4+ and CD8+ can be explored more thoroughly as a screening for disease status and can be an useful and consistent predictor of COVID-19 severity.

### Funding
The authors received no funding for this work.

### Competing Interests
The authors declare that they have no competing interests.

### Author Contributions
- Phey Liana conceived and designed the experiments, performed the experiments, analyzed the data, prepared figures and/or tables, authored or reviewed drafts of the article, and approved the final draft.
- Aprilia Paskah Samosir conceived and designed the experiments, performed the experiments, analyzed the data, prepared figures and/or tables, and approved the final draft.
- Nurmalia Purnama Sari conceived and designed the experiments, analyzed the data, authored or reviewed drafts of the article, and approved the final draft.
- Raden Ayu Linda Andriani conceived and designed the experiments, authored or reviewed drafts of the article, and approved the final draft.
- Verdiansah Verdiansah conceived and designed the experiments, authored or reviewed drafts of the article, and approved the final draft.
- Hidayatullah Hidayatullah conceived and designed the experiments, authored or reviewed drafts of the article, and approved the final draft.

- Zen Ahmad conceived and designed the experiments, authored or reviewed drafts of the article, and approved the final draft.
- Tungki Pratama Umar conceived and designed the experiments, analyzed the data, prepared figures and/or tables, authored or reviewed drafts of the article, and approved the final draft.

## Human Ethics

The following information was supplied relating to ethical approvals (*i.e.*, approving body and any reference numbers):

The Sriwijaya University Faculty of Medicine Ethics Committee approved the study procedure (Approval number: 212-2022).

## Data Availability

The raw measurements are available in the Supplemental File.

## Supplemental Information

Supplemental information for this article can be found online at http://dx.doi.org/10.7717/peerj.15509#supplemental-information.

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
