# Peer review of "CD4+ and CD8+ cell counts are significantly correlated with absolute lymphocyte count in hospitalized COVID-19 patients: a retrospective study"

_PeerJ, doi:10.7717/peerj.15509_

## Round 0.1 · original submission · Major Revisions

The manuscript has been assessed by three independent reviewers and I strongly suggest addressing the concerns raised by all three reviewers before your paper could be considered for publication.

1. Language revision is highly recommended for better understanding.
2. The rationale for the study needs to be consistent throughout the manuscript, discrepancy is observed between the abstract and the discussion.
3. The figures need extensive reformatting. Authors are recommended to maintain uniformity in data representation.
4. The authors are highly recommended to provide a detailed explanation for the statistical method used.

Reviewer 1 ·

Basic reporting

The authors should improve the overall writing style with regards to english grammar and sentence structure.

The introduction is well structured, however I wold recommend providing better explanation of key terms use such as "Lymphopenia". Along these lines it's unclear to me what is meant with 'laboratory disturbances' (line 51) and 'laboratory impairment' (line 151). Additionally, what is meant with 'this process' in line 55?

I would highly recommend to provide additional references particularly in line 61, 69.

It's essential for the overall understanding of the manuscript to provide figure legends describing the data sown in figure 1 and 2.

Can you please provide the raw data in a commonly used format such as xls or pdf or doc? The current raw data is provided as .sav, which didn't allow me to open the file.

The overall structure of the manuscript is fine. I would recommends to use a common format of the manuscript. It seem that the discussion has a different line spacing compared to the rest of manuscript.

Experimental design

The research question of the abstract seems different from the research question discussed in the results and discussion part in the actual manuscript. Specially, the abstract states that the aim of the study is 'to examine the correlation between CD4+ and CD8+ cell counts and absolute lymphocyte count (ALC) in COVID-19 patients'. However in the results, discussion and conclusion the authors highlight the finding that lymphocyte subsets were significantly reduced in severe COVID-19 as compared to mild COVID-19 patients. I would recommitment to better phrase the research question.

The data table provided gives a nice overview of the results obtained. To strengthen the findings obtain, I would recommend to proved additional data plots. For example, it seem interesting that lymphocyte counts change between mind and severe COVID-10 patients. Can you provide plots individually for CD8, CD4 and NK cells?

Validity of the findings

Can you please provide the rational of comparing patients at 'admission' vs '10th day'?
Why did you choose 10 days as a secondary time point?
Are patients 10 days after hospital admission virus free?
Are you following up with the the same patients 10 days after they were admitted to the hospital?

Can you please provide the raw data for all you plots?

Since T cells are the major component of the lymphocyte population, isn't it expected that counts are correlated? What is rational that numbers would not correlate in COVID-19 patients?

Additional comments

no comment

Reviewer 2 ·

Basic reporting

In this study, Liana et al., aims to analyze the correlation between CD4+ and CD8+ cell counts and ALC in COVID-19 patients from March 2022 to May 2022.

Experimental design

The experimental observations are not significantly different in examined groups and do not add any valuable information to existing knowledge.

Validity of the findings

The data show no significant difference in claimed immune cells populations among ALCs. The data set is incomplete and no scientific conclusion can be made based on the presented data.

·

Basic reporting

The manuscript by Liana and coworkers titled “CD4+ and CD8+ cell count is significantly correlated with absolute lymphocyte count in hospitalized COVID-19 patients” describes correlation between absolute lymphocyte counts and CD4+ and CD8+ cell counts in COVID19 patients in a small set of patients from Indonesia. The study was designed well, and manuscript was written well. Sample size is small, but it seems to be enough to detect the statistical associations. Results were adequately supported by figures. This manuscript requires minor corrections.
1) I ask authors to check their figure and make sure that they have all the essential components such as axis labels, and axis titles. The Y axis label of figure 1 could be cells/mm3. The Y and the X axes labels of all panels in figure 2 could show units along with the cell types.
2) The phrase ‘abnormal data distribution’ seems incorrect statistically. You may state that ‘the data do not follow normal distribution’ instead.
3) Please correct sentence in line 94 and 95. You may want to write ‘Wilcoxon signed rant test’ instead of ‘Wilcoxon test’ for paired data!
4) How did you test normality of data distribution? Did you also check difference in variance while deciding between t test and Mann Whitney U test? Please add this information in the methods section.

Experimental design

No comment.

Validity of the findings

No comment.

---

## Round 0.2 · accepted · Accept

The authors have adequately addressed the comments and the manuscript could be accepted in the present form.

Reviewer 1 ·

Basic reporting

The publication by Liana and colleagues meets the reporting structure.

I would recommend adding the figure legend below the figure not above.
In line 128 and 140, I would remove the strikethrough.

Experimental design

The publication by Liana and colleagues meets the experimental design

Validity of the findings

The author's have improved and addressed all comments raised during the first revision.

·

Basic reporting

My comments were addressed.

Experimental design

None

Validity of the findings

None